# Small-scale farming in drylands: New models for resilient practices of millet and sorghum cultivation

**Abel Ruiz-Giralt**[1], **Stefano Biagetti**[1,2], **Marco Madella**[1,2,3], **Carla Lancelotti**[1,3]*

**1** CaSEs Research Group, Department of Humanities, Universitat Pompeu Fabra, Barcelona, Spain,
**2** School of Geography, Archaeology and Environmental Studies (GAES), University of the Witwatersrand, Johannesburg, South Africa, **3** ICREA, Barcelona, Spain

* carla.lancelotti@upf.edu

## Abstract

Finger millet, pearl millet and sorghum are amongst the most important drought-tolerant crops worldwide. They constitute primary staple crops in drylands, where their production is known to date back over 5000 years ago. Compared to other crops, millets and sorghum have received less attention until very recently, and their production has been progressively reduced in the last 50 years. Here, we present new models that focus on the ecological factors driving finger millet, pearl millet and sorghum traditional cultivation, with a global perspective. The interaction between environment and traditional agrosystems was investigated by Redundancy Analysis of published literature and tested against novel ethnographic data. Contrary to earlier beliefs, our models show that the total annual precipitation is not the most determinant factor in shaping millet and sorghum agriculture. Instead, our results point to the importance of other variables such as the duration of the plant growing cycle, soil water-holding capacity or soil nutrient availability. This highlights the potential of finger millet, pearl millet and sorghum traditional cultivation practices as a response to recent increase of aridity levels worldwide. Ultimately, these practices can play a pivotal role for resilience and sustainability of dryland agriculture.

## Introduction

Finger millet (*Eleusine coracana* Gaertn.), pearl millet (*Pennisetum glaucum* (L.) R.Br.) and sorghum (*Sorghum bicolor* (L.) Moench) are amongst the most important drought tolerant crops in the world. These cereals are cultivated in several ecological regions, but are most common in drylands, where they constitute primary food crops [1–3]. Compared to other crops, such as wheat (*Triticum* ssp.), maize (*Zea mays* L.) or rice (*Oryza* ssp.), sorghum and millets require less water input during their growth and therefore can be cultivated in areas with water deficit. A minimum of 300 mm/yr for millets and 350/400 mm/yr for sorghum are considered necessary for the development of seeds [4]. This entails that in all those areas where annual rainfall is lower than 300 mm, especially during the period of plant growth, it would not be possible to cultivate these crops without irrigation. However, there are examples of

**Data Availability Statement:** Primary research data and R code used for the analyses are stored on Zenodo at https://doi.org/10.5281/zenodo.6501883.

**Funding:** This research was funded by the European Research Council with a Starting Grant awarded to CL (ERC-Stg 759800). The funders had no role in study design, data collection and analysis, decision to publish, or preparation of the manuscript.

**Competing interests:** The authors have declared that no competing interests exist.

modern communities that do cultivate these crops extensively, under exclusively rainfed conditions, in areas where annual average precipitation is much lower [5, 6]. This inconsistency between academic and traditional knowledge has been also highlighted in recent work by the Ceres2030 consortium (https://ceres2030.org/), which identified a significant mismatch between research on solutions to world hunger and the needs of small-scale farmers [7]. In this paper, we aim at:

1. analysing the extent of sorghum, finger millet and pearl millet cultivation in areas with limited rainfall;

2. understanding how people engage with a practice that is supposedly not viable in drylands;

3. exploring the ecological drivers behind the cultivation of sorghum, finger and pearl millet;

We approach this investigation through an ethnographic and cross-cultural modelling perspective. Differently from the yield-oriented models normally developed in agronomic studies [8–13], we focus on the decision-making mechanisms behind the choice of growing finger millet, pearl millet and sorghum, as well as on the techniques that have been traditionally applied to cultivate such cereals, regardless of production outputs. Our working-hypothesis is that models that include TEK are able to better predict current agricultural practices in drylands than those that do not take it into account. This information holds enormous value in the current search towards ecological sustainability [14] and food security [15] as it results from extremely resilient social-ecological systems, which have been in place for extended periods of time and are a consequence of long-term processes of ecological adaptation [16, 17]. We integrate traditional ecological knowledge (TEK) with academic ecological knowledge (AEK) to create models that aim to understand how traditional agricultural systems relate to their surrounding environment. TEK is also referred to as local or indigenous ecological knowledge (LEK, IEK); local knowledge is defined as the knowledge of a particular community living in a specific location because of traditional, external and contemporary learning; indigenous knowledge refers to culturally embedded explanations of reality; and traditional knowledge contemplates the part of local knowledge that is transmitted through generations [18]. AEK is also referred to as scientific or Western ecological knowledge (SEK, WEK). Ludwig and Poliseli [19] argue for the use of AEK to avoid conflicts generated by the concept of what is scientific (SEK would imply that LEK, IEK and AEK have no scientific base) or the provenance of the scientists (WEK suggest that only Western knowledge can be considered academic) [19].

Traditional agricultural practices are mainly determined by the combined effect of plant growth rhythms and the surrounding environment [20]. Even though agricultural activities can be related to several factors, such as market economy, technological implementations or social-cultural tradition that contribute to the high variability of agricultural systems [21], we consider that, under specific environmental and cultural contexts, societies can only adopt a finite number of agricultural solutions. In this work we concentrate on the ecological drivers rather than the cultural background of cultivation practices. As so, we designed a model that analyses the cultivation and farming techniques of rural communities with non-market economies (TEK data), in relation to crop characteristics and environmental data in which they are applied (AEK data). For this purpose, we created a database of published and novel ethnographic data on all the known communities that cultivate one or more of the target crops, independently to their environmental, ecological, or technological background. We included also communities living in humid areas to capture all the variability of conditions in which these three crops are grown. However, in the discussion we concentrate on drylands as sorghum and millets are sometimes the only crops available and constitute a staple food whereas

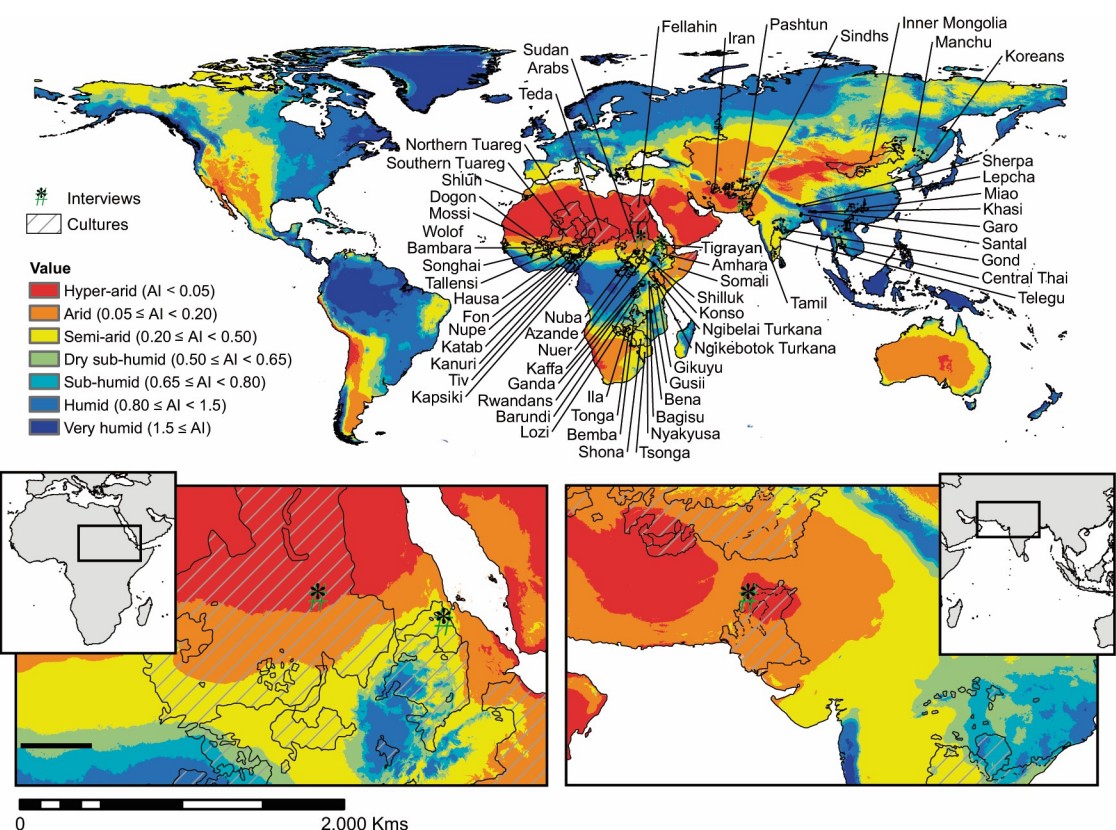

**Fig 1. World's regions classification according to aridity index values, territorial distribution of ethnographic groups (eHRAF [23]) as indicated by GREG polygons [24], and location of ethnographic interviews.**

in humid environments these species represent one of the many that are cultivated and usually have a subsidiary role.

Drylands are generally defined by the scarcity of water, which affects the environment and its natural resources, and therefore determines and drives human economic activities. The United Nations Environment Programme (UNEP) has provided a clear definition of drylands according to an aridity index (AI), expressing the ratio between average annual precipitation and potential evapotranspiration [22]. According to the UNEP, drylands are lands with an AI <0.65, and can be further divided, into hyper-arid (AI <0.05), arid (0.05–0.2), semi-arid (0.2–0.5) and dry sub-humid (0.5–0.65) lands. Drylands represent 41% of the global land area and are to be found throughout all continents (Fig 1).

Characterized by patchy and limited resources, often ephemeral and erratic, drylands–especially hyper-arid to arid–are generally seen as 'marginal' areas for human settlement and food production. They are often over-exploited ecosystems, where minor shifts in rainfall can trigger drastic changes in the environment, which can in turn ignite episodes of drought, famine, and migrations [25]. Nonetheless, drylands have seen the emergence and the development of many urban entities throughout the last six millennia, like Ancient Egypt, Mesopotamian states and empires, the Indus Valley Civilisation, the kingdom of Aksum, the Zimbabwe cultures, or the Mesoamerican states, among others. These large-scale processes (e.g., emergence of state) occurred close to rivers or better watered areas (e.g., the dry sub-humid zones). Only recently the role of drylands at large has been revaluated and considered by some scholars as

active centres of innovations throughout history [26, 27]. The legacy of long-term past adaptations is nowadays to be found in the traditional ecological knowledge of current drylands inhabitants, who developed through time a variety of innovative solutions to produce food under strong environmental constraints. Yet in many reconstructions of agricultural land use and system productivity, large portions of drylands (hyper-arid and arid, see 1) are considered almost totally unproductive, under the assumption that below a given rainfall cultivation is not viable [28]. This holds true even when the data refers to rainfed cultivation of drought-resistant and drought-tolerant species, such as millets and sorghum, and both global [29] and regional viewpoints [30] do not consider arid and hyper-arid lands as suitable areas for cultivation. The reason for this might partially reside in that most of these maps are generated using a combination of production statistics, land use data, satellite imagery and biophysical characteristics [31].

Traditional knowledge has often been considered irrational rather than sustainable, under biased perspectives funded on typically western concepts [32]. As previously stressed by Krätli and colleagues [33], development plans have often failed to provide long-term support and rehabilitation to drylands communities in case of drought or famine. As a result, overall socioeconomic conditions in drylands are far worse than in other parts of the planet, and not surprisingly world poverty is concentrated in drylands [22]. Recently, a number of papers have questioned the 'traditional' approach to drylands and have explored different perspectives. Largely inspired by New Ecology [34, 35] and the adoption of the concept of resilience of social-ecological systems (as formulated by Holling [36] and applied with success to drylands [37–40]), TEK is being considered a possible way to design sustainable and durable approaches in agroecology [41]. We present here the results of ethnographic and cross-cultural investigations on the cultivation of pearl millet, finger millet and sorghum, with a global perspective (1). We use published ethnographic material (see the Material and Methods section as well as Supplementary Information and SI Dataset S1) and novel data collected on the field (Datasets SI2 and SI3) to build and test models that display the interaction of ecological and geographic variables in explaining agricultural practices in drylands.

## Methods

### Ethnographic data: Systematic interviews and eHRAF

Traditional Ecological Knowledge on the cultivation practices of sorghum, pearl millet and finger millet was extracted from both primary and secondary sources. On the one hand, ethnographic fieldwork was carried out in Tigray (Ethiopia), Khartoum State (Sudan) and Sindh (Pakistan) in 2018 and 2019 [42], during which several interviews were conducted with people engaged in traditional agricultural practices (Fig S1 in S1 Appendix). Oral consent was obtained and recorded prior to the interview from each participant as approved by the Institutional Committee for Ethical Review of Projects (CIREP) at Universitat Pompeu Fabra (ethics certificate n. 2017/7662/I). All methods were carried out in accordance with relevant guidelines and regulations. A total of 53 semi-structured interviews, which were systematically completed using a questionnaire as a general guide, provided the data for testing the model performance. The questions targeted data on agricultural activity related to finger millet, pearl millet and sorghum production, including information about cereal species selection and cultivation, farming methods and techniques, water management practices, growing cycles, land tenure, alimentation, and food-security, amongst other topics. Participants were selected through snowball sampling, always under the advice and approval of local authorities and colleagues. Interviewees were predominantly landowners from rural and semi-rural areas, ranging between 27 and 88 years old and whose main economic activity was farming. They

included 46 men and 7 women, which had been farmers for the most part of their lives. The total number of interviews performed in each area depended on the availability of participants in a radius of less than 100 km from the base camp. Retrieved information was processed, normalized, and added into a separated dataset both as single entries for each interview but also as aggregated data for each of the three cultures: Tigrayan, Sudan Arabs and Sindhis. Detailed results of these interview can be found in Biagetti et al. [42].

Data from existing anthropological studies were obtained from the Human Relation Area Files (eHRAF) [43]. The objective was to provide a consistent, coherent body of information, which allowed for the creation of a robust dataset for cross-cultural comparison on agricultural activities in drylands. The eHRAF database allows to perform comparative studies [23] by providing easy access to a wide range of ethnographic sources and it is being increasingly used to carry out ethnoarchaeologically driven pre-search [44, 45]. The inclusion criteria for extracting the information from eHRAF for this study were:

1. Cultivation of one or more of the target crops, independently to the environmental, ecological, or technological background,

2. Being small-scale food producers,

3. Database entries contained explicit information on crops, and cultivation techniques.

In the present study, we included all occurrences reporting the cultivation of finger millet (FM), pearl millet (PM) and/or sorghum (SB). The study variables taken into account included: intensity of cultivation (casual, extensive and intensive), watering regimes (rain-fed, décrue and irrigation) and the duration of the growing cycle of each crop (Table S1 in S1 Appendix). All ethnographic bibliography containing both generic and specific terms referring to the three crops under study was extracted from eHRAF and systematically reviewed (Table S3 in S1 Appendix). Data was separated and organized by community for a total of 66 entries. This preliminary dataset was normalized into a cultures database which included pre-created categories on socio-economic features (e.g., type of subsistence economy, settlement, or group mobility) and plant cultivation practices and techniques (e.g., crop importance, cycle duration, land preparation, manuring or watering systems) based on the Standard Cross-Cultural Sample Codebook [46] (Table 1). Further to eHRAF data, available bibliography was reviewed in order to fill missing information in the database and only the communities with

**Table 1. Definitions of agricultural practices considered in this study.**

| Variable | Definition |
|---|---|
| Casual agriculture | Slight or sporadic cultivation of food or other plants incidental to a primary dependence upon other subsistence practice [47] |
| Extensive agriculture | Or shifting cultivation, as where new fields are cleared annually, cultivated for a year or two, and then allowed to revert to forest or brush for a long fallow period [47] |
| Intensive agriculture | On permanent fields, utilizing fertilization by compost or animal manure, crop rotation, or other techniques so that fallowing is either unnecessary or is confined to relatively short periods [47] |
| Rain-fed agriculture | Water is provided by rainfall alone (directly or as run-off), cultivation occurs far from any permanent water sources and without any water harvesting [5] |
| *Décrue* agriculture | Water is provided by natural inundation, typically from major river systems (floodplain cultivation) [5] |
| Irrigated agriculture | Water is provided to crops at regular intervals throughout the growing season by human intervention [5] |
| Duration of growing cycle | Mean and variance of crops' growing cycle duration (in days) from sowing to harvest [23] |

enough information to define all the current study variables were retained for analysis, resulting in a final dataset of 57 entries. Finally, as this research concentrates on agricultural techniques rather than on social aspects, the database was organized by crop growing cycles taking into account that some societies cultivate in each year 2 crops with different techniques, hence reaching a total of 72 entries.

## Environmental data and spatial distribution

A total of 58 ecological variables both physio-climatic and edaphic were included, as they are considered to be the principal factors in plant growth and development [48]. Environmental data were extracted from published GIS data at 30 arc-secs resolution and derived raster files created with ArcGIS 10.6 or QGIS 3.4.15 with GRASS 7.8.2 (Table S3 in S1 Appendix). Mean values and variances for each ecological variable were included in the analysis, resulting in a grand total of 116 variables. Data retrieval was based on previously assigned "areas of activity" (Fig 1): the cultures spatial distribution was obtained from the Geo-Referencing of Ethnic Groups dataset (GREG [49]), which employs geographic information systems (GIS) to represent group territories as polygons independently of state boundaries. In case of no data, the location of societies was assigned by using their administrative units as described in eHRAF documents. Territories designated to each culture were not restricted to agriculturally active areas but included their whole area of activity. Centroids of these polygons were utilized in order to define longitude and latitude for each human community. Furthermore, the geographic location of the ethnographic interviews was established as a 50-kilometer-round area from the GPS location of each subject house. This choice was based on the information about agricultural fields location given during the interviews, which ranged between 0 and 40 kilometers. All operations were performed using R 3.6.2, specifically the rgdal [50], raster [51], and spatialEco [52] packages.

## Data analysis and modelling

The eHRAF data was used as training response variables, whereas the ethnographic dataset was utilized as testing response data. Both datasets were transformed into dummy binary variables and divided into 4 subsets for separate analysis:

1. presence or absence of each study crop (all cases, n = 72)

2. agricultural intensity and watering systems for finger millet (n = 30)

3. agricultural intensity and watering systems for pearl millet (n = 27)

4. agricultural intensity and watering systems for sorghum (n = 55)

Redundancy analysis (RDA [53]) was applied in order to analyse each response subset variability in relation to the duration of the plants growing cycle and their surrounding environment. All training response datasets were transformed using Hellinger's transformation prior to RDA [54, 55], whereas the explanatory datasets were standardized (by subtracting the variable mean to each value and then dividing it for the standard deviation) to create comparable scales. First, RDA was applied to explore the overall variability of each subset of data, accounted for by growing cycle and environmental predictors. The proportion of inertia retained by each of these components was also retrieved as the adjusted coefficient of determination ($R2$ [56]). Permutation tests were used to check for statistical significance of each RDA [57] and the variance inflation factors of each variable (VIF [58] cited in [53]) were calculated to look for linear dependencies between explanatory variables. Second, adjusted-$R2$-based forward selection (FS [59]) was used to identify and select significant predictor variables and

reduce collinearity—as it can work with supersaturated models [60]. A double-stopping criterion (alpha level combined with the adjusted R2) was implemented and tested over 1000 permutations [59]. The resulting models were analysed for explained inertia and statistical significance, as were the FS variables for collinearity and statistical relevance. Model coefficients for each FS predictor and ordination scores for both response variables and study cases were calculated in order to understand the effect of each explanatory variable in the response data.

Next, variation partitioning (VP) was performed to test for spatially structured variance [56, 59]. For this purpose, models were used along with XY coordinates and distance-based Moran's eigenvector maps (dbMEMs [53, 60, 61] but see also [62, 63]). Linear trends of each response data subset were analysed by RDA following Borcard et al. [64]. When statistically significant, response data was de-trended prior to dbMEM analysis by regressing all response variables on the XY coordinates and retaining the residuals [53]. The construction of dbMEMs [60, 64] was carried out using minimum distance between polygon frontiers as geographical distances amongst study cases. RDA was then applied for each response data subset against their dbMEMs. The resulting spatial submodels were tested for statistical significance and FS was applied when confirmed by 1000 permutations. VP analysis [64] was used to decompose the total inertia into independent and shared fractions: that is, the pure fraction of each explanatory dataset; their joint fractions as a result of intercorrelation, and the remaining unexplained variation. Testable shared fractions were evaluated by RDA, whereas the pure individual fractions of each predictor dataset were tested by means of partial RDA. Models were evaluated using performance measures: accuracy (correctly classified entries / total number of cases), recall (positive entries correctly classified / total number of positive cases), precision (positive samples that were correctly classified / total number of positive predicted cases) and F1-score (evaluation of the classification performance through calculation of the harmonic mean of precision and recall [65]). A classification threshold was obtained by using the sensitivity-specificity sum maximization approach on the training data [66–69]. The models were then validated by assessing their effectiveness on predicting their training datasets. Next, accuracy and F1-score were measured when predicting the testing response data. Fig S2 in S1 Appendix presents a summary of all the described methods, whereas Fig S3 in S1 Appendix shows the full schematic workflow of the analysis. All statistical analyses were executed using R 3.6.2, specifically the FactoMineR [70], factoextra [71], vegan [72], rgeos [73], adespatial [74], and PresenceAbsence [75] packages.

## Results

### Crop selection and cultivation practices

Table 2 presents descriptive statistics of crop selection and cultivation training response data. Finger millet cultivation practices were identified by cross tabulation as Extensive-Rainfed (56.6%), Intensive-Rainfed (36.7%) and Intensive-Irrigated agriculture (6.7%). The results of cross tabulations for pearl millet cultivation were similar to that of finger millet, but with a higher presence of irrigated systems: 55.6% entries were classified as Extensive-Rainfed, 29.6% as Intensive-Rainfed and 14.8% as Intensive-Irrigated agriculture. Sorghum agriculture featured a higher rate of diversity: along with Extensive-Rainfed (40%), Intensive-Rainfed (36.4%) and Intensive-Irrigated (10.9%), two additional groups were identified by cross tabulation as Casual-Rainfed (7.3%) and Intensive-Décrue (5.4%), for a total of five combinations for sorghum cultivation. In no instance, casual agriculture was observed to be combined with décrue or irrigated watering regimes, neither was extensive agriculture.

**Table 2. Descriptive statistics of training response data.** (n) absolute number, (f) frequency.

| Study crops | | Finger millet—FM | | Pearl millet—PM | | Sorghum—SB | |
|---|---|---|---|---|---|---|---|
| **Attribute** | **n** | **f** | **%** | **f** | **%** | **f** | **%** |
| **Crop selection** | 72 | 30 | 41.6 | 27 | 37.5 | 55 | 76.3 |
| **Intensity of cultivation** | 72 | 30 | | 27 | | 55 | |
| Casual agriculture (CAS) | | 0 | 0.0 | 0 | 0.0 | 4 | 7.3 |
| Extensive agriculture (EXT) | | 17 | 56.7 | 15 | 55.6 | 22 | 40.0 |
| Intensive agriculture (INT) | | 13 | 43.3 | 12 | 44.4 | 29 | 52.7 |
| **Watering regimes** | 72 | 30 | | 27 | | 55 | |
| Rainfed agriculture (RF) | | 28 | 93.3 | 23 | 85.2 | 46 | 83.7 |
| D´ecrue agriculture (DEC) | | 0 | 0.0 | 0 | 0.0 | 3 | 5.4 |
| Irrigated agriculture (IRR) | | 2 | 6.7 | 4 | 14.8 | 6 | 10.9 |

## Modelling variability of traditional cultivation practices

After FS, RDA showed the models to retain 54.9% of the total inertia for crop selection and 60.7% for finger millet, 87.8% for pearl millet and 24%, for sorghum cultivation. All selected variables were found to be statistically significant and independent to one another. For crop selection, six variables appeared as the most relevant (SI Figure 4a in S1 Appendix): mean topsoil volumetric water content at 15 kPa, mean topsoil pH, variance of mean temperature of the warmest quarter, mean global horizontal irradiance, variance of subsoil clay content and mean precipitation seasonality. Significant variables for finger millet cultivation include mean subsoil sulphur content, mean precipitation concentration index and topsoil mean phosphorus content (SI Figure 4b in S1 Appendix). The most relevant variables for pearl millet cultivation were variance of temperature seasonality, variance of topsoil volumetric water content at 33 kPa, mean subsoil gravel content, mean topsoil clay content, mean duration of the growing cycle, the mean temperature of wettest quarter, variance of topsoil organic carbon content, variance of topsoil silt content and mean temperature during the driest quarter (SI Figure 4c in S1 Appendix). The most important variables for sorghum cultivation were the mean of growing cycle duration, the variance of both topsoil and subsoil cation exchange capacity and the mean soil organic carbon (SI Figure 4d in S1 Appendix).

## Absence of spatial patterns

Linear trend analysis by RDA revealed statistically significant models for crop selection and pearl millet cultivation variables. None of the analyses performed with distance-based Moran's eigenvector maps (dbMEMs) were found to be statistically significant, nor was any dbMEM selected by means of FS, hence pointing to the absence of spatial autocorrelation in both finger millet and sorghum cultivation datasets. As a result, dbMEMs were not included in variation partitioning analysis (VP). For crop selection, VP results (Fig 2a) showed significant effects of physio-climatic, edaphic, and spatial components on the variability of the study agricultural package (19.3%, 39.5% and 17.8% of the total inertia). 12% of the variance retained by edaphic factors was also explained by the spatial component, thereby pointing to the existence of a linear trend amongst edaphic variables. Still, the pure spatial fraction failed to pass the test for statistical significance, hence pointing to the absence of spatial patterns in the crop selection dataset. VP analysis of finger millet cultivation data identified the impact of both physio-climatic and edaphic components to be statistically significant. No shared fraction was identified (Fig 2b). For pearl millet cultivation, a component related to the duration of the plant growing cycle (7.9%) was also detected along with the physio-climatic (32.8%), edaphic (27.3%) and

### a) Crop selection

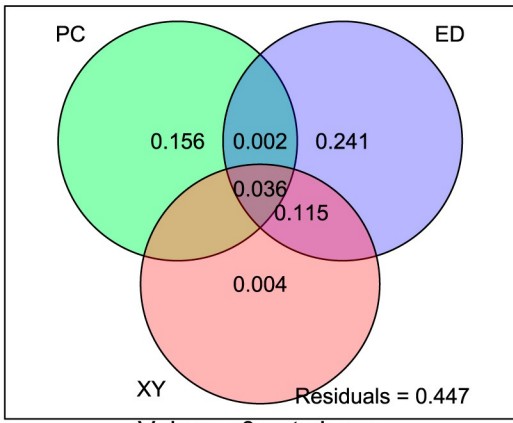

### b) FM cultivation

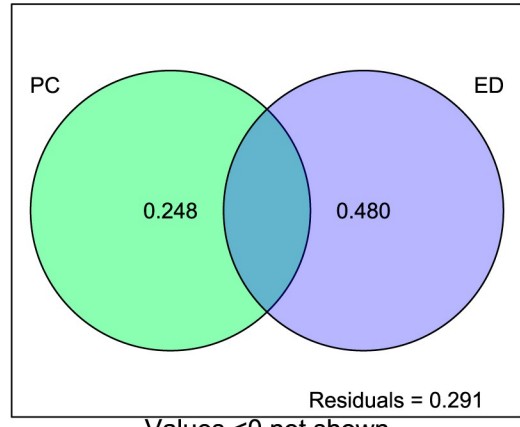

### c) PM cultivation

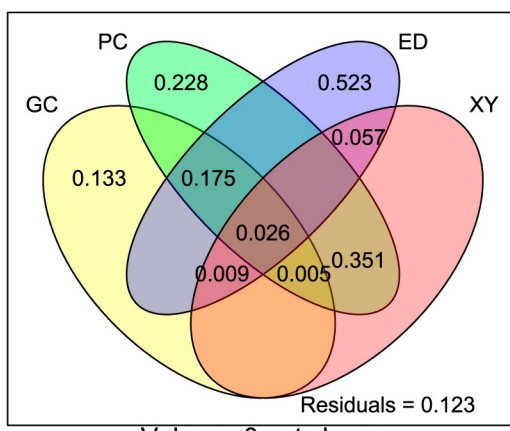

### d) SB cultivation

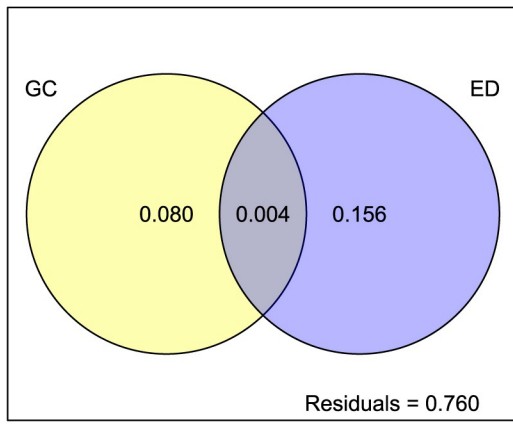

**Fig 2.** Summary by Venn diagrams of VP analysis of physio-climatic (PC), edaphic (ED), spatial (XY) and plant's growing cycle (GC) components of a) Crop selection; b) Finger millet (FM) cultivation; c) Pearl millet (PM) cultivation; and d) Sorghum (SB) cultivation.

spatial components (25.8%)—all of which were found to be statistically significant. The pure fraction of the spatial component was proven to retain no inertia hence showing the absence of spatial autocorrelation in the pearl millet dataset—even though the shared fraction with the rest of the components points to the existence of linear spatial trends amongst the predictors (Fig 2c). As for the unique contributions of each component, they were all found to be statistically meaningful. Finally, both growing cycle and edaphic components were found to significantly explain 8.4% and 16% of the total variability in sorghum cultivation (Fig 2d).

## Model validation using ethnographic observations

All four models were found to be capable of predicting their own training response datasets (Fig S5 in S1 Appendix). The crop selection model showed 86.6% accuracy and a F1-score of 0.869, with precision and recall featuring values of 0.88 and 0.857 respectively. 95% accuracy was obtained for the finger millet cultivation model, whereas the prediction of the pearl millet training data was 100% accurate. All the performance measures scored 0.95 and 1 respectively.

Finally, the modelling of sorghum cultivation practices showed 78.2% accuracy and a F1-score of 0.723. In this case, recall was found to be larger (0.855) than precision (0.627) indicating a higher rate of false positives amongst the predictions. All models scored between 60% and 80% accuracy when predicting individual cases (Fig S6a in S1 Appendix). Interestingly, the models F1-score (Fig S6b in S1 Appendix) remained similar to accuracy for crop selection, as well as for finger millet and pearl millet cultivation models. However, the sorghum cultivation model classification strength (F1-score) was lower than its accuracy by 8% due to a higher rate of false positives (0.4) than false negatives (0.233). Regarding the prediction of the testing cases as cultures (individuals mode), the models showed an accuracy of 77.8% for crop selection, 100% for finger millet, 50% for pearl millet and 83.3% for sorghum (Fig S6c in S1 Appendix). Again, F1-scores (Fig S6d in S1 Appendix) for crop selection, finger millet and pearl millet cultivation models featured almost no change with respect to accuracy, whereas the sorghum cultivation model also showed lower precision (0.714) than recall (0.833).

## Discussion

Traditional agricultural systems have been receiving enhanced attention [14, 16, 17, 21, 76, 77], especially after the introduction of FAO's climate-smart agriculture initiative in 2010 [78]. However, the integration of traditional practices into institutionalised science and policy seems to be still marginal [16]. Far from being static, traditional practices are constantly hybridizing with both local and global knowledge, thereby hindering the task of creating comprehensive datasets susceptible to ecological analysis. As a result, parameters such as the annual rainfall limit are still used to assess the cultivation suitability of a given area [24, 28, 79, 80]. This produces a general mismatch between research on drought-resistant cultivations in drylands and the reality of many small-scale farmers in these areas [7]. Water scarcity is generally considered one of the main limiting factors to agricultural production. Even for drought-resistant crops such as finger millet, pearl millet and sorghum, mean annual precipitation is generally regarded as the critical factor that defines agroecological systems. However, our model on crop selection portrays a different picture showing that total yearly rainfall, although important, does not appear to be as critical as previously suggested. Indeed, mean annual precipitation is not retained as a variable in the model and it only explains 8.3% of the overall variance when used as the only predictor in the crop selection model. Finger millet is preferred by groups inhabiting areas with higher soil water-retention capacity (e.g., Shilluk, Gusii, Bagisu), whereas pearl millet is chosen by communities living in areas where seasonal precipitations are more temporally concentrated, such as the Fellahin, Wolof or Dogon.

More importantly, water availability is not the only driving factor in either case. The selection of finger millet is further associated with areas with higher regional variance of summer temperature (e.g., Pashtun, Sherpa, Amhara), indicating the plant capacity to resist greater intra-regional temperature ranges. Indeed, finger millet has been recognized as a high-temperature tolerant species, with landraces resisting over 40˚C [81]. Enhanced solar irradiance (more commonly insolation when reported integrated over a time period) determines the choice of pearl millet by communities such as the Teda, Kanuri or southern Tuareg. This characteristic of pearl millet makes it very suitable to areas such as the Sahara and its margins as the species has high efficiency in converting solar radiation into dry matter, especially in comparison with C3 crops [82]. The inclusion of sorghum in traditional agrosystems appears to be unrelated to water availability and associated with relatively higher soil water pH, present in areas such as Inner Mongolia, Somalia or the Turkana region. Indeed, soil acidity has been found to significantly reduce sorghum yields [83]. Recent research has shown that pH increases with aridity and temperature [84], suggesting that sorghum might be part of the

drylands crop package for being able to cope with alkalinity induced by aridity. Overall, our crop selection model indicates that in traditional agricultural systems this choice is highly influenced by ecological conditions.

Our models explained a significant part of the total variability of cultivation practices, especially for finger and pearl millet. All three models performed well when cross validated against our first-hand ethnographic data. This is especially notable if we consider the impact of current technological implementations such as tractor agriculture or water-pumping techniques and the effects of state policies on land tenure and availability, but also social factors such as the influence of globalization on individuals' preferences or beliefs about agricultural productivity. Precipitation concentrated in a short period of the year was found to be associated with extensive-rainfed regimes of finger millet cultivation. These types of agrosystems are traditionally developed by human communities occupying regions with high rainfall seasonality, both in sub-humid to very humid areas (e.g., Azande, Khasi or Garo) and drylands with AI <0.40 (e.g., Nuba, Shilluk or Tonga). Notably, this was the only rain-related predictor to have a significant impact in all three models. In our finger millet model, intensive-rainfed systems are connected with regions characterized by high topsoil phosphorus content (e.g., Konso, Nyakyusa, Kaffa). Plant-available soil phosphorus has been identified as a crucial factor for sorghum and experimental cultivation has shown that sorghum and finger millet respond similarly to P [85]. Finally, high subsoil sulphur concentrations seem to be a driver for irrigation as the only two instances of recorded irrigated finger millet (e.g., Pashtun and Tamil) are strongly related to this variable. To our knowledge, no study has yet investigated the relation of sulphur and finger millet watering practices.

By contrast, pearl millet cultivation systems featured growing cycle, edaphic and temperature-related predictors as their significant ecological driving factors. Previous studies have argued for water stress as the main limitation to pearl millet cultivation [82, 86–88] despite having identified the importance of other factors such as soil nutrient availability [89]. According to our model, extensive-rainfed cultivation is preferred by farmers planting slow-growing pearl millet varieties in lands with high topsoil clay content (e. g. Fon, Ila, Shona), which allows for better water retention regardless of aridity (AI ranging from 0.14 to 1.04). By contrast, communities such as the Mossi, Nupe or Songhai developed intensive-rainfed systems in arid to dry sub-humid areas where plant-available soil water was much more irregularly distributed as a result of enhanced soil water loss—due to increased evaporation and drainage. Irrigated agrosystems were developed in both hyper-arid (e.g., southern Tuareg, Teda) and semi-arid (e.g., Telugu) environments where soil water evaporation processes were even more significant; but also, more irregularly distributed at an intra-regional scale.

Rainfall was not found to play a direct role in traditional techniques applied to sorghum cultivation either. Instead, our sorghum cultivation model indicate that the most crucial factors were related to the duration of the growing cycles as well as to soil fertility variables, in accordance with previous reports [4]. According to our results, supplementary growing cycles appeared in relation to rainfed cultivation regimes developed by communities such as the Azande, Santal or Tiv. In these cases, the main limitation to intensive cultivation is soil fertility regardless of aridity (AI between 0.14 to 1.04 for extensive-rainfed cultivation, and 0.16 to 1.07 for intensive-rainfed agriculturalists). Certainly, higher concentrations of soil organic matter allowed for the implementation of intensive cultivation systems (e.g., Koreans, Rwandans, Gikuyu), whereas communities in less fertile regions such as the Hausa, Bambara or Wolof have to use extensive or land-shifting regimes. Communities living in hyper-arid to arid areas where fertility is unevenly distributed and concentrated around water sources, showed application of décrue and irrigated watering practices (e.g., Fellahin, Shluh or southern Tuareg people). Still, communities living in more humid areas such as Central

Thais and Tamils were also found to use irrigation. By contrast, casual-rainfed sorghum production appeared restricted to hyper-arid to arid regions where reduced soil organic matter paired with higher intra-regional variability of topsoil cation exchange capacity (e.g., the areas around water sources).

Overall, our models reveal the existence of important ecological patterns in the ways that traditional small-scale farmers adapt to their surrounding environment, most of which showed no direct relationship with annual rainfall nor aridity levels. Variation partitioning analysis detected no variability exclusively driven by geographical location or distance between communities. As so, we argue that processes of cultural transmission did not play a primary role in the shaping of the studied agrosystems, which were instead the result of local processes of adaptation [44]. The existing similarities can thus be considered as a product of cultural convergence, as several communities reached similar agroecological solutions when faced with similar ecological problems independently of cultural diversity. As so, traditional agricultural knowledge appears as a type of TEK resulting from long-term adaptation processes [90, 91] that allowed for the development of sustainable, resilient agroecosystems. Most of the main driving ecological factors described in the present study were found to be in agreement with previous academic ecological knowledge.

## Concluding remarks

The ecological modelling of traditional agricultural systems has revealed that the relationship between annual precipitation and agricultural viability is not as strong as previously considered. Other factors such as growing cycles duration, soil nutrient availability and water holding capacity appear to be much more determinant in shaping traditional agroecosystems. Our work forwards the understanding of how human communities developed long-term sustainable, resilient agricultural strategies. This is especially significant in the current context of climate instability and increasing population, which calls for immediate action. Global climate change is fostering new research on local practices and traditional crops. TEK offers a highly relevant source of information, as it encompasses the exploitation of locally available resources, and it is the result of long-term processes of adaptation to the environment. By contrast, supra-national institutions have often opted for short-term, generalized solutions such as the so-called improvement of the seed market with high-yielding hybrids or the promotion of agrochemicals in economically less developed regions. Despite their relatively positive short-term effect on crop yields, these solutions are based on finite resources, and have caused significant damage to both crop biodiversity and soil conservation. Instead, traditional practices rely mainly on renewable resources, and they can be considered as a suitable way to increase productivity and minimize crop failure without sacrificing sustainability and resilience on the long-term scale. Besides, and in parallel to the improvement of crops to increase drought tolerance and yield, the current situation calls for the revaluation of small-scale agricultural strategies suited to specific agroecosystems. The present study offers an alternative view on possible pathways to integrate traditional knowledge in scientific and policy programs to provide solutions to food security for low-and middle-income dryland areas.

## Supporting information

**S1 Appendix. Extended results, figures and tables document containing supplementary information: Extended results, figures, and tables.**
(PDF)

## Acknowledgments

We are grateful for the collaboration with: the Eastern Tigray Archaeological Project (ETAP) and for the participation of the Authority for Research and Conservation of Cultural Heritage (ARCCH) and the Tigray Tourism and Cultural Commission(TCTB); El Salha project—supported by the Italian Ministry for Foreign Affairs–under the auspices of the National Corporation for Antiquities and Museums (Sudan); Shah Abdul Latif University in Khairpur, Pakistan, the Director General for Archaeology of Sindh and the chief police officer in Qambar Shahdadkot District (Sindh). Finally, many thanks are due to the residents of the Gulo Makeda district (Tigray, Ethiopia), and Khartoum state (Khartoum, Sudan), as well as to the inhabitants of the Dadu, Qambar Shahdadkot and Larkhana districts (Sindh, Pakistan) for their time, patience and hospitality. We also acknowledge the invaluable help of Mongeda Khalid Magzoub, Yamane Meresa, Mulubrhan Haile Gebreselassie, Ghulam Mohiuddin Veesar, Tasleem Alam Abro, and Amin Chandio in conducting the ethnographic interviews. All authors are members of CaSEs Research Group of the Catalan Research Agency (AGAUR SGR-212), which is an associated unit (Unidad Asociada) to the Institución Milà y Fontanals of the Spanish Research National Council (CSIC).

## Author Contributions

**Conceptualization:** Stefano Biagetti, Marco Madella, Carla Lancelotti.

**Formal analysis:** Abel Ruiz-Giralt.

**Funding acquisition:** Carla Lancelotti.

**Investigation:** Abel Ruiz-Giralt, Stefano Biagetti, Carla Lancelotti.

**Project administration:** Carla Lancelotti.

**Resources:** Abel Ruiz-Giralt.

**Supervision:** Stefano Biagetti, Carla Lancelotti.

**Visualization:** Abel Ruiz-Giralt.

**Writing – original draft:** Abel Ruiz-Giralt, Stefano Biagetti, Marco Madella, Carla Lancelotti.

**Writing – review & editing:** Abel Ruiz-Giralt, Stefano Biagetti, Marco Madella, Carla Lancelotti.

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
