## [Decision Letter · Decision Letter 0]

7 Jul 2021

PONE-D-21-06936

Small-scale farming in drylands: New models for resilient practices of millet and sorghum cultivation

PLOS ONE

Dear Dr. Lancelotti,

Thank you for submitting your manuscript to PLOS ONE. After careful consideration, we have decided that your manuscript does not meet our criteria for publication and must therefore be rejected.

I am sorry that we cannot be more positive on this occasion, but hope that you appreciate the reasons for this decision.

Yours sincerely,

Rattan Singh Yadav, PhD

Academic Editor

PLOS ONE

Additional Editor Comments (if provided):

Please provide additional data as suggested by reviewer 2

Reviewers' comments:

Reviewer's Responses to Questions

**Comments to the Author**

1. Is the manuscript technically sound, and do the data support the conclusions?

Reviewer #1: No

Reviewer #2: Partly

2. Has the statistical analysis been performed appropriately and rigorously? 

Reviewer #1: I Don't Know

Reviewer #2: I Don't Know

3. Have the authors made all data underlying the findings in their manuscript fully available?

Reviewer #1: Yes

Reviewer #2: Yes

4. Is the manuscript presented in an intelligible fashion and written in standard English?

Reviewer #1: Yes

Reviewer #2: Yes

5. Review Comments to the Author

Reviewer #1: My problem with this paper lies in the fact that I cannot discern why the work was done. The Introduction doesn’t describe a problem to be addressed nor a hypothesis that could be tested. The results themselves are poorly described – what is the function of Fig 1 and what does Fig 2 represent in real terms? Consequently, the Discussion and Conclusions also fail to reveal what all the work was about, other than to make another plea for TEK to be taken into account in agricultural research and development. I have no problem with that at all but this paper doesn’t add anything useful to support that plea.

At the outset the authors acknowledge that it is not just the ecological and geographical variables that determine crop choice by farmers, but then go on to exclude any consideration of socio-economic, markets, taste preferences, cultural or historical issues. There are very many reasons why farmers grow the crops they do and some reasons may have nothing to do with how well the crops are suited to the physical conditions. A classic case is that of farmers in the drier regions of Southern Africa (Botswana and southern Zimbabwe) who know that maize will fail often but plant it every year anyway (often citing, amongst other reasons, the labour savings in birdscaring to protect the ‘naked’ seeds of sorghum and pearl millet). In a related issue, the data used in this paper do not appear (I may be wrong) to include information on whether the crops chosen actually grow very well. Apart from many other possible reasons for choosing them it may be that other crops fare even worse. Or they used to grow well historically but the conditions have worsened? I could go on and on.

I do not believe this manuscript is worthy of publication in PLOS One.

Reviewer #2: Unable to assess the analysis. The text is well written and clear. Nothing more to addUnable to assess the analysis. The text is well written and clear. Nothing more to addUnable to assess the analysis.

6. PLOS authors have the option to publish the peer review history of their article (what does this mean?). If published, this will include your full peer review and any attached files.

Reviewer #1: No

Reviewer #2: No

- - - - -

---

## [Author Response · Author response to Decision Letter 0]

19 Oct 2021

Additional Editor Comments (if provided):

Please provide additional data as suggested by reviewer 2

RE. Reviewer 2 does not ask for additional data, so we are not sure what this comment refers to. If the editor refers to reviewer #1’s comments, please see our answers below.

Reviewers' comments:

Reviewer's Responses to Questions

Comments to the Author

1. Is the manuscript technically sound, and do the data support the conclusions?

Reviewer #1: No

Reviewer #2: Partly

RE. We do not understand how the reviewers have reached this decision considering that they both state in their review, and in response to question 2 below, that they are not able to assess the analysis or evaluate the data. Consequently, in our opinion, they should not be able to evaluate whether the data support the conclusion or if the experiments (in this case modelling experiments) have been conducted rigorously.

2. Has the statistical analysis been performed appropriately and rigorously? 

Reviewer #1: I Don't Know

Reviewer #2: I Don't Know

RE. Considering that this is a modelling paper heavily based on statistics, this answer alone should disqualify the reviews as experts who can provide informed assessments.

5. Review Comments to the Author

Reviewer #1:

- My problem with this paper lies in the fact that I cannot discern why the work was done. The Introduction doesn’t describe a problem to be addressed nor a hypothesis that could be tested.

RE. We appreciate the fact that the aims of the papers might not be sufficiently clear and we are happy to clarify them if needed. However, we state what is the purpose of the paper in the introduction. Copied from Introduction:

Our work contributes to this debate [i.e. the recognised inconsistency between academic and traditional knowledge in agricultural policies] by providing a global assessment of, and novel data on traditional practices adopted by small-scale farmers. Differently from the yield-oriented models normally developed in agronomic studies, we focus on the techniques that have been traditionally applied to cultivate these crops by small-scale farmers. We integrate traditional ecological knowledge (TEK) with academic ecological knowledge (AEK) in order to create models that aim at understanding how traditional agricultural systems relate to the surrounding environment. (Introduction, page 2 lines 12-19)

- The results themselves are poorly described – what is the function of Fig 1 and what does Fig 2 represent in real terms?

RE. Figure 1 is a map showing the location of the data points. Figure 2 is a Venn diagram showing variation partition analysis results. Clearly the reviewer cannot understand the figures (although Figure 1 being a map should be pretty self-explanatory) as he/she does not understand the analysis that is at the core of the whole paper. We also provide extended results in the appendix, with much detail on the statistical analysis, which the reviewer has probably not even looked at considering his/her inability to understand the statistics.

- Consequently, the Discussion and Conclusions also fail to reveal what all the work was about, other than to make another plea for TEK to be taken into account in agricultural research and development. I have no problem with that at all but this paper doesn’t add anything useful to support that plea.

RE. Again, if the reviewer does not understand the paper, he/she cannot understand whether the conclusions are sound and novel.

- At the outset the authors acknowledge that it is not just the ecological and geographical variables that determine crop choice by farmers, but then go on to exclude any consideration of socio-economic, markets, taste preferences, cultural or historical issues.

RE. In all modelling papers some variables are excluded from the analysis. The choice of which variables to retain and which to exclude is fully justified in the paper:

Even though agricultural activities can be influenced by several factors, […]-which all contribute to the high variability of agricultural systems [16]- we consider that, under specific environmental and cultural contexts, societies can only adopt a finite number of agricultural solutions. However, as it has been shown that ecological or medical reasons are often at the root of food preferences, even in their more extreme form of food taboos [17], in this work we concentrate on the ecological drivers rather than the cultural background of cultivation practices. (Introduction, pages 3-3. Lines 25-33)

- There are very many reasons why farmers grow the crops they do and some reasons may have nothing to do with how well the crops are suited to the physical conditions. A classic case is that of farmers in the drier regions of Southern Africa (Botswana and southern Zimbabwe) who know that maize will fail often but plant it every year anyway (often citing, amongst other reasons, the labour savings in birdscaring to protect the ‘naked’ seeds of sorghum and pearl millet). In a related issue, the data used in this paper do not appear (I may be wrong) to include information on whether the crops chosen actually grow very well. Apart from many other possible reasons for choosing them it may be that other crops fare even worse. Or they used to grow well historically but the conditions have worsened? I could go on and on.

RE. Indeed, the list of reasons why people decide to cultivate some crops and not others is probably limitless. That is why, when using a model-based approach to try and pinpoint the general mechanisms behind these decisions, some variables (such as cultural decisions) cannot be taken into consideration.

- I do not believe this manuscript is worthy of publication in PLOS One.

RE. In our opinion, and considering what has been detailed above, no sound reason has been given on why this decision was reached, especially considering the poor understanding of the modelling approach used in this paper.

Reviewer #2:

- Unable to assess the analysis. The text is well written and clear. Nothing more to addUnable to assess the analysis. The text is well written and clear. Nothing more to addUnable to assess the analysis.

RE. We do not really have any response to this reviewer rather than reiterate the fact that if he/she is unable to assess the analysis, he/she is not qualified to be a reviewer of this paper. Nonetheless we appreciate the fact that he/she considers the paper well written and clear.

---

## [Decision Letter · Decision Letter 1]

11 Mar 2022

PONE-D-21-06936R1Small-scale farming in drylands: New models for resilient practices of millet and sorghum cultivationPLOS ONE

Dear Dr. Lancelotti,

Thank you for submitting your manuscript to PLOS ONE. After careful consideration, we feel that it has merit but does not fully meet PLOS ONE’s publication criteria as it currently stands. Therefore, we invite you to submit a revised version of the manuscript that addresses the points raised during the review process.

ACADEMIC EDITOR:I am sorry for a delayed decision. It was hard to find the reviewers. One reviewer have now commented on your manuscript. Based on the comments and my own expertise, I think the manuscript can be published in Plos One. However, several changes are needed to bring the manuscript in a better form. The introduction must provide a sound rationale of using new models and the deficiencies in existing models. The MM has skipped several information that how interviews were conducted and how the selection was made. I suggest to carefully revise the manuscript keeping in view the comments made by the reviewer.

We look forward to receiving your revised manuscript.

Kind regards,

Shahid Farooq, Ph.D.

Academic Editor

PLOS ONE

Journal Requirements:

"This research has been developed in the framework of the Resilience and Adaptation to Drylands: Identifying past water management practices for drought-resistant crops (RAINDROPS) project funded by the European Research Council (ERC-Stg2017 G.A.759800). We are grateful for the collaboration with: the Eastern Tigray ArchaeologicaProject (ETAP) - funded by the Social Sciences and Humanities Research Council of Canada (Partnership Grant n. 890-215-003) - as well as for the participation of the Authority for Research and Conservation of Cultural Heritage (ARCCH) and the Tigray Tourism and Cultural Commission (TCTB); El Salha project - funded by the Italian Ministry for Foreign Affairs - under the auspices of the National Corporation for Antiquities and Museums (Sudan); Shah Abdul Latif University in Khairpur, Pakistan, the Director General for Archaeology of Sindh and the chief police officer in Qambar Shahdadkot District (Sindh). Finally, many thanks are due to the residents of the Gulo Makeda district (Tigray, Ethiopia), and Khartoum state (Khartoum, Sudan), as well as to the inhabitants of the Dadu, Qambar Shahdadkot and Larkhana districts (Sindh, Pakistan) for their time, patience and hospitality. All authors are members of CaSEs Research Group of the Catalan Research Agency (AGAUR SGR-212), which is an associated unit (Unidad Asociada) to the Instituci´on Mil`a y Fontanals of the Spanish Research National Council (CSIC)."

"This research was funded by the European Research Council with a Starting Grant awarded to CL (ERC-Stg 759800). The funders had no role in study design, data collection and analysis, decision to publish, or preparation of the manuscript."

4. We note that Figures 1 and S1 in your submission contain map/satellite images which may be copyrighted. All PLOS content is published under the Creative Commons Attribution License (CC BY 4.0), which means that the manuscript, images, and Supporting Information files will be freely available online, and any third party is permitted to access, download, copy, distribute, and use these materials in any way, even commercially, with proper attribution. For these reasons, we cannot publish previously copyrighted maps or satellite images created using proprietary data, such as Google software (Google Maps, Street View, and Earth). For more information, see our copyright guidelines: http://journals.plos.org/plosone/s/licenses-and-copyright.

a) You may seek permission from the original copyright holder of Figures 1 and S1 to publish the content specifically under the CC BY 4.0 license.

5. We note that your manuscript is not formatted using one of PLOS ONE’s accepted file types. Please reattach your manuscript as one of the following file types: .doc, .docx, .rtf, or .tex (accompanied by a .pdf).

If your submission was prepared in LaTex, please submit your manuscript file in PDF format and attach your .tex file as “other.”

6. We note that the grant information you provided in the ‘Funding Information’ and ‘Financial Disclosure’ sections do not match.

Additional Editor Comments (if provided):

Reviewers' comments:

Reviewer's Responses to Questions

**Comments to the Author**

1. If the authors have adequately addressed your comments raised in a previous round of review and you feel that this manuscript is now acceptable for publication, you may indicate that here to bypass the “Comments to the Author” section, enter your conflict of interest statement in the “Confidential to Editor” section, and submit your "Accept" recommendation.

Reviewer #3: All comments have been addressed

2. Is the manuscript technically sound, and do the data support the conclusions?

Reviewer #3: Yes

3. Has the statistical analysis been performed appropriately and rigorously? 

Reviewer #3: Yes

4. Have the authors made all data underlying the findings in their manuscript fully available?

Reviewer #3: Yes

5. Is the manuscript presented in an intelligible fashion and written in standard English?

Reviewer #3: Yes

6. Review Comments to the Author

Reviewer #3: I have evaluated the manuscript, “Small-scale farming in drylands: New models for resilient practices of millet and sorghum cultivation”. The authors combined published and novel data on traditional cultivation of finger millet, pearl millet and sorghum, with a global perspective. The authors concluded that our models show that the total amount of rainfall per year is not determining the viability of cultivation and identify the most relevant variables explaining farming variability. The manuscript adds significant knowledge to the field. However, several modifications are needed to bring the manuscript in publishable form. The recommendations are as under;

i. The abstract is very raw. Nothing has been provided in the abstract that how models were built and which. Abstract must contain all necessary information. The rainfall could not be the only variable affecting the production of different crops. Therefore, exclude this information from the abstract section.

ii. The language of the manuscript needs a careful checking. There are plenty of long sentences which must be split in order to improve overall readability.

iii. TEK and AEK abbreviations are used for different terms (lines 21-22 and 39-40)

iv. Lines 46 to onward belong to MM along with table, please shift these there

v. The length of introduction is so small and the relevant literature has been skipped. The most of the introduction tells the story what authors have done. It must contain a sound rationale that why new models are necessary. Further the difference from earlier (in use) models and advantages of using these models must be provided in the introduction section. There must be some hypothesis/objectives given at the end of introduction section.

vi. There is no author in the list from Pakistan. How the interviews were conducted in Sindh?

vii. Each section in the results starts from relevant methods. Please delete all methods-related text from the results

viii. Concluding remarks must not be based on the earlier published studies. Just provide some key take-home messages.

ix. The manuscript can be accepted after these changes.

7. PLOS authors have the option to publish the peer review history of their article (what does this mean?). If published, this will include your full peer review and any attached files.

Reviewer #3: No

---

## [Author Response · Author response to Decision Letter 1]

29 Mar 2022

Review Comments to the Author

We have addressed all the comments provided by the reviewer. Please see point-by-point answers below:

i. The abstract is very raw. Nothing has been provided in the abstract that how models were built and which. Abstract must contain all necessary information. The rainfall could not be the only variable affecting the production of different crops. Therefore, exclude this information from the abstract section.

Response: the abstract has been rewritten to add information on how we built our models and what are the principal results, and it now reads as follow:

Finger millet, pearl millet and sorghum are amongst the most important drought-tolerant crops worldwide. They constitute primary staple crops in drylands, where their production is known to date back over 5000 years ago. Compared to other crops, millets and sorghum have received less attention until very recently, and their production has been progressively reduced in the last 50 years. Here, we present new models that focus on the ecological factors driving finger millet, pearl millet and sorghum traditional cultivation, with a global perspective. The interaction between environment and traditional agrosystems was investigated by Redundancy Analysis of published literature and tested against novel ethnographic data. Contrary to earlier beliefs, our models show that the total annual precipitation is not the most determinant factor in shaping millet and sorghum agriculture. Instead, our results point to the importance of other variables such as the duration of the plant growing cycle, soil water-holding capacity or soil nutrient availability. This highlights the potential of finger millet, pearl millet and sorghum traditional cultivation practices as a response to recent increase of aridity levels worldwide. Ultimately, these practices can play a pivotal role for resilience and sustainability of dryland agriculture.

ii. The language of the manuscript needs a careful checking. There are plenty of long sentences which must be split in order to improve overall readability.

Response: the manuscript has been thoroughly revised and all long sentences simplified. We hope that it now reads better. 

iii. TEK and AEK abbreviations are used for different terms (lines 21-22 and 39-40)

Response: We use the acronyms TEK and AEK with the same meaning throughout the text. In lines 21-22 (lines 32-24 in the new manuscript) we give the definitions of the two acronyms whereas in lines 39-40 (lines 44-45 in the new manuscript) we provide examples of what are considered TEK and AEK data in the present work. For clarity we have added ‘data’ after the acronyms in lines 39-40.

iv. Lines 46 to onward belong to MM along with table, please shift these there

Response: The section has been incorporated into the Material and Method (line 122 and following in the new manuscript). 

v. The length of introduction is so small and the relevant literature has been skipped. The most of the introduction tells the story what authors have done. It must contain a sound rationale that why new models are necessary. Further the difference from earlier (in use) models and advantages of using these models must be provided in the introduction section. There must be some hypothesis/objectives given at the end of introduction section.

Response: We have substantially improved the introduction by adding a section on agriculture in drylands (lines 54 and following) that includes a review of current models used to map agricultural productivity and what we consider to be their shortcomings (lines 79-93) and why we consider that TEK should be integrated in these maps (hence why our models add relevant information to the current practice, lines 93-104).

Our working-hypothesis is now specified in lines 30-32 of revised manuscript

vi. There is no author in the list from Pakistan. How the interviews were conducted in Sindh?

Response: The interviews were conducted with the invaluable help of local collaborators who are all authors of the paper that describes in detail the results of the interviews (Biagetti et al in press, citation 42 in the revised manuscript). They are not authors on this paper as they have not directly participated in the building of the models. Notwithstanding, we realise we should have at least acknowledged them in the Acknowledgement section. We have now added their names to it and we thank the reviewer for pointing this out. 

vii. Each section in the results starts from relevant methods. Please delete all methods-related text from the results

Response: all the text related to methods was removed form the results unless it was deemed necessary to the understanding of the results themselves.

viii. Concluding remarks must not be based on the earlier published studies. Just provide some key take-home messages.

Response: the concluding remarks have been shortened and made more to the point. All references have been deleted.

ix. The manuscript can be accepted after these changes.

Response: we thank the reviewer for their useful comments.

---

## [Editor Report · Decision Letter 2]

14 Apr 2022

PONE-D-21-06936R2

Small-scale farming in drylands: New models for resilient practices of millet and sorghum cultivation

PLOS ONE

Dear Dr. Lancelotti,

Thank you for submitting your manuscript to PLOS ONE. After careful consideration, we feel that it has merit but does not fully meet PLOS ONE’s publication criteria as it currently stands. Therefore, we invite you to submit a revised version of the manuscript that addresses the points raised during the review process.

Thank you for the careful revision of the manuscript. I have noticed some minor errors and your manuscript will be accepted after these revisions.Scientific name of maize is spelled incorrectly in introduction line 6.The table legends must be above the tables, not below.The figures must be incorporated within the text, like tables.Agriculture in drylands heading should be deleted. This gives the impression that the paper is a review article.In Table 1, the first letter of second word, i.e., agriculture must be written small caps in column 1The n and f in Table 2 must be explained in footnotes  The scientific names of all species in the references section must be italic.

We look forward to receiving your revised manuscript.

Kind regards,

Shahid Farooq, Ph.D.

Academic Editor

PLOS ONE
---

## [Author Response · Author response to Decision Letter 2]

19 Apr 2022

Please, find below a point-by-point response to the latest comments.

• Scientific name of maize is spelled incorrectly in introduction line 6. – This has been corrected

• The table legends must be above the tables, not below. - This has been corrected

• The figures must be incorporated within the text, like tables. – Figures have been added (to add them to the LaTeX we had to use the .png and .pdf versions of the files. The files uploaded separately in the previous round are exactly the same just in .tiff version)

• Agriculture in drylands heading should be deleted. This gives the impression that the paper is a review article. – The heading has been deleted

• In Table 1, the first letter of second word, i.e., agriculture must be written small caps in column 1 – This has been changed

• The n and f in Table 2 must be explained in footnotes - This has been added to the caption

• The scientific names of all species in the references section must be italic. – This has been corrected (the change is not reflected in the manuscript with differences highlighted as the bibliography is a separate LaTeX file that is not checked when producing the diff.tex file)

---

## [Editor Report · Decision Letter 3]

25 Apr 2022

Small-scale farming in drylands: New models for resilient practices of millet and sorghum cultivation

PONE-D-21-06936R3

Dear Dr. Lancelotti,

We’re pleased to inform you that your manuscript has been judged scientifically suitable for publication and will be formally accepted for publication once it meets all outstanding technical requirements.

Kind regards,

Shahid Farooq, Ph.D.

Academic Editor

PLOS ONE
---

## [Editor Report · Acceptance letter]

13 May 2022

PONE-D-21-06936R3 

Small-scale farming in drylands: New models for resilient practices of millet and sorghum cultivation 

Dear Dr. Lancelotti:

I'm pleased to inform you that your manuscript has been deemed suitable for publication in PLOS ONE. Congratulations! Your manuscript is now with our production department. 

Kind regards, 

on behalf of

Dr. Shahid Farooq 

%CORR_ED_EDITOR_ROLE%

PLOS ONE